# Abscisic Acid Synthesis and Signaling during the Ripening of Raspberry (*Rubus idaeus* ‘Heritage’) Fruit

**DOI:** 10.3390/plants12091882

**Published:** 2023-05-05

**Authors:** Fernanda Álvarez, Mario Moya, Claudia Rivera-Mora, Paz E. Zúñiga, Karla Jara-Cornejo, Paula Muñoz, Aníbal Ayala-Raso, Sergi Munné-Bosch, Carlos R. Figueroa, Nicolás E. Figueroa, Mónika Valdenegro, Juan E. Alvaro, Wilfried Schwab, Bruno G. Defilippi, Lida Fuentes

**Affiliations:** 1Centro Regional de Estudios en Alimentos Saludables (CREAS), CONICYT-Regional GORE Valparaíso Proyecto R17A10001, Avenida Universidad 330, Placilla, Curauma, Valparaíso 2362696, Chile; falvarez@creas.cl; 2Laboratory of Plant Molecular Physiology, Institute of Biological Sciences, Universidad de Talca, Talca 3465548, Chile; mariomoyamo@gmail.com (M.M.); claudia.rivera@utalca.cl (C.R.-M.); paz.zuniga@utalca.cl (P.E.Z.); karla.jara@utalca.cl (K.J.-C.); cfigueroa@utalca.cl (C.R.F.); 3Millennium Nucleus for the Development of Super Adaptable Plants (MN-SAP), Santiago 8340755, Chile; 4Departament de Biologia Evolutiva, Ecología i Ciències Ambientals, Facultat de Biologia, Universitat de Barcelona, Avinguda Diagonal, 645, E-08028 Barcelona, Spain; paula.munoz@ub.edu (P.M.); smunne@ub.edu (S.M.-B.); 5Instituto de Estadística, Facultad de Ciencias, Universidad de Valparaíso, Valparaíso 2360102, Chile; anibal.ayala@postgrado.uv.cl; 6Biotechnology of Natural Products, Technical University Munich, D-85354 Freising, Germany; ibv.nicfigueroa@gmail.com (N.E.F.); wilfried.schwab@tum.de (W.S.); 7Escuela de Agronomía, Facultad de Ciencias Agronómicas y de los Alimentos, Pontificia Universidad Católica de Valparaíso, Casilla 4-D, Quillota 2260000, Chile; monika.valdenegro@pucv.cl (M.V.); juan-eugenio.alvaro@pucv.cl (J.E.A.); 8Unidad de Postcosecha, INIA La Platina, Santa Rosa, Santiago 8820000, Chile; bdefilip@inia.cl

**Keywords:** ABA biosynthesis and signaling, ethylene, *NCED* and *PYL* genes, fruit quality parameters, *Rubus idaeus*, softening

## Abstract

The raspberry (*Rubus idaeus* L.) fruit is characterized by its richness in functional molecules and high nutritional value, but the high rate of fruit softening limits its quality during postharvest. Raspberry drupelets have a particular ripening regulation, depending partially on the effect of ethylene produced from the receptacle. However, the possible role of abscisic acid (ABA) in the modulation of quality parameters during the ripening of raspberry is unclear. This study characterized the fruit quality-associated parameters and hormonal contents during fruit development in two seasons. The quality parameters showed typical changes during ripening: a drastic loss of firmness, increase in soluble solids content, loss of acidity, and turning to a red color from the large green stage to fully ripe fruit in both seasons. A significant increase in the ABA content was observed during the ripening of drupelets and receptacles, with the higher content in the receptacle of ripe and overripe stages compared to the large green stage. Moreover, identification of ABA biosynthesis-(9-cis-epoxycarotenoid dioxygenase/NCED) and ABA receptor-related genes (PYRs-like receptors) showed three genes encoding RiNCEDs and nine genes for RiPYLs. The expression level of these genes increased from the large green stage to the full-ripe stage, specifically characterized by a higher expression of *RiNCED1* in the receptacle tissue. This study reports a consistent concomitant increase in the ABA content and the expression of *RiNCED1, RiPYL1,* and *RiPYL8* during the ripening of the raspberry fruit, thus supporting the role for ABA signaling in drupelets.

## 1. Introduction

Raspberries have a unique flavor, an attractive color, and valuable health benefits (rich source of minerals and vitamins, dietary fiber, and various antioxidants) [1,2]. However, raspberries are soft fruit with a fast ripening rate, leading to a short postharvest shelf life. In addition, it has been reported that fruit ripening in raspberry is characterized by a progressive decrease in the firmness of the fruit, associated with cell wall modification and expression of cell wall degrading-related genes; starting from the large green to overripe stages of ripening [1,3,4,5,6]. 

To date, little is known about the hormonal regulation of raspberry ripening and its impact on the final quality of this fruit. Raspberry is categorized as a non-climacteric fruit [7], although ethylene production and respiratory peaks have been detected in the receptacle from the white fruit stage until full maturity [5,8,9]. Therefore, ethylene has been reported to partially regulate firmness loss during postharvest, but this regulation is temperature-dependent [9]. In addition, this hormone would be responsible for the detachment of the drupes to the receptacle [10]. Also, the indole-3-acetic acid (IAA) conjugation with amino acids might play a central role in the onset of raspberry ripening by decreasing the IAA-free content [11]. Recently, it has been reported that the maximum accumulation of abscisic acid (ABA) takes place at the ripe stage [12]. The above finding is in agreement with our previous de-novo assembly analysis during different fruit developmental stages, which showed differentially expressed genes (DEGs) that included transcripts related to the synthesis and response to ABA; content regulation, as well as the influx and efflux of auxin; response to auxin; and ethylene biosynthesis and perception [13]. 

In grape and strawberry fruits, it has been pointed out that ABA is the main regulator of maturation, its biosynthesis is necessary for the onset of ripening, and its increase is concomitantly with the progression of maturity [14,15,16]. In strawberry, ABA treatment was reported to accelerate fruit softening and increase color and ethylene production [17,18]. Also, the role of genes encoding enzymes related to ABA metabolism such as 9-cis-epoxycarotenoid dioxygenase (NCED) and abscisic acid 8’-hydroxylase 1-like in the grape and strawberry ripening has been demonstrated [15,19,20,21]. A remarkable decrease in the ABA content and a significantly retarded ripening process resulted from the down-regulation of *FaNCED1* [15]. In turn, the pyrabactin resistance 1 (PYR1)/PYR1-like (PYL) is an ABA receptor core of ABA signaling components [20,22,23]. The PYL family members have been described in climacteric and non-climacteric fruit, increasing the expression of these genes during the ripening of the fruits of tomato [24], strawberry [22,25], and grape [26]. 

Despite our understanding of the fundamental physiological and biochemical aspects of non-climacteric fruit such as raspberry, many molecular mechanisms involved in the raspberry fruit development and ripening, as well as their effect on quality parameters, still need to be fully understood. In this study, we observed typical changes in quality attributes during the ripening of raspberry in two seasons. We showed that ABA content increases in parallel to raspberry ripening in both drupe and receptacle tissues. In addition, ABA-related genes were identified, and marked differences were observed in the expression profile of the *RiNCED* and ABA receptor genes (*RiPYL*s) depending on the fruit tissue. 

## 2. Results

### 2.1. Fruit Quality Measurements

In this study, the fruit of raspberry (*Rubus idaeus* ‘Heritage’) was classified into five developmental stages [9,27] during the 2020 and 2021 harvest seasons in Casablanca, Valparaíso Region, Chile (33°20′39″ S; 71°22′11″ W; 247 masl) (Figure 1). 

During the 2020 season, firmness decreased 40% from the W to P stages. Conversely, in the 2021 season, the firmness drastically decreased 48% from the LG to W stages (Figure 2). A significant increase in total soluble solids (TSS) was observed from the LG to W stages and the W to P stages during both seasons, with a larger TSS content during the 2020 season compared to the 2021 season (Figure 2). A sharp decrease in titratable acidity (TA) was observed from the LG to W stages (Figure 2), with a higher acidity content at the LG stage of the 2021 season; however, the acidity of ripe stages was similar in both seasons. 

### 2.2. ABA and ACC Dynamics during Raspberry Fruit Development

In this study, the contents of abscisic acid (ABA) and 1-amino-cyclopropane-1-carboxylic acid (ACC) were determined by ultrahigh-performance liquid chromatography coupled to electrospray ionization tandem mass spectrometry (UPLC/ESI-MS/MS). 

The ABA content was determined in drupelets and receptacles of different fruit developmental stages and during two seasons (Figure 3). During the development of the raspberry fruit, the ABA content increased in both tissues and seasons at the R and OR stages compared to the LG stage (Figure 3). Furthermore, in both seasons, the ABA content was higher (10,000–20,000 ng/gDW) in receptacles than in drupelets (6000–7500 ng/gDW) at the R stage (Figure 3).

Previously, we showed the role of ethylene in partially regulating the firmness of raspberry in a temperature-dependent manner during the postharvest season [9]. Therefore, the content of ACC, the immediate ethylene precursor, was determined during ripening in both seasons. The ACC content showed significant differences in receptacles between the LG and R stages in the 2021 season only (Figure 3).

To determine the relationship of the ABA and ACC contents with the parameters of the fruits’ quality, a correlation matrix analysis was performed between variables (Figure 4). In the 2020 season, the standardized components describe principally a positive correlation between the variables determined in drupelets, i.e., the TSS and ABA content (0.62), and a negative correlation between firmness and ABA content (−0.66) and between the TA and ABA content (−0.49) (Figure 4). In the 2021 season, a positive correlation was observed between the TSS and ABA content (0.48), and negative correlations were observed between firmness and the ABA content (−0.66), and between the TA and ABA content (−0.61) in drupelets (Figure 4).

### 2.3. Identification of RiNCED and RiPYL Genes in the Raspberry Genome

Since ABA increases significantly during ripening (P, R, and OR stages. Figure 3), we characterized the genes related to the ABA biosynthesis-(9-cis-epoxycarotenoid dioxygenase-like/*RiNCED*) and ABA receptor-related genes (PYRs-like receptors/*RiPYL*) identified in the draft genome published by Wight et al., 2019 [28]. The study of gene families for each gene in the raspberry genome shows three and nine members for the *RiNCED* and *RiPYL* genes, respectively (Appendix A). The *RiNCED* sequence lengths ranged from 2005 to 2670 bp, and the deduced protein lengths were from 604 to 614 amino acid residues (Appendix A). However, the gene sequence lengths of the *RiPYL* gene family ranged from 750 to 2973 bp, and the deduced protein length varied between 160 to 216 aminoacidic residues (Appendix A).

### 2.4. Structure, Conserved Motifs and Promoter Analysis of RiNCED and RiPYL Genes

The gene structure and conserved motifs of the *RiNCED* and *RiPYL* genes and deduced proteins identified in *Rubus idaeus* were explored using the AUGUSTUS Server and MEME Suite, respectively (Figure 5 and Figure 6). The gene structure and motif analysis showed a high degree of conservation in the *RiNCED* gene family. The three *RiNCED* sequences did not show introns (Figure 5A). The results showed that the three RiNCED-deduced proteins exhibited six highly conserved motifs (Figure 5B). Each RiNCED-deduced protein showed six conserved motifs similarly distributed in the three proteins identified, indicating the low variability in this gene family. 

However, the analysis of nine sequences of the *RiPYL* gene family showed that *RiPYL1-6* and *RiPYL 9* have no introns; conversely, three introns were found for *RiPYL7* and two introns for the *RiPYL8* gene (Figure 6A). The results showed that the nine RiPYL deduced proteins exhibited three similarly distributed conserved motifs (Figure 6B) indicating the low variability in the final protein sequence of this gene family. 

The analysis of the promoter regions of the *RiNCED* and *RiPYL* genes for the identification of cis-elements for the phytohormone response, such as ABA, auxin, ethylene, gibberellins, jasmonates, and salicylic acid was performed using the draft genome of raspberry [28]. The analyses showed that most *RiNCED* and *RiPYL* sequences present regulatory elements for ABA (Figure 7). Thus, *RiNCED3* and *RiPYL2*, *RiNCED1*, and *RiPYL7* and *RiPYL9* show 15, nine, and six ABA response cis-elements, respectively. It is also shown that these promoter regions present cis-elements of the response to other hormones, observing six ethylene cis-elements for the *RiPYL5* promoter sequence. In addition, most of the identified sequences showed cis-elements in the response to gibberellin (except for *RiNCED1*), jasmonates (except for *RiPYL8* and *RiPYL9*), and salicylic acid (except for *RiPYL9*) (Figure 7).

### 2.5. Phylogenetic Analysis and Classification of RiNCED and RiPYL Deduced Proteins

The phylogenetic analysis showed that each RiNCED family member belongs to the three subfamilies—group I, II, and V—of the five NCED groups described in the plants (Figure 8). RiPYL proteins were distributed within the three groups described in the plants (Figure 8). The deduced protein sequences show a length of 614, 604, and 613 amino acid residues for RiNCED1, RiNCED2, and RiNCED3, respectively (Appendix A). The RiNCED1 predicted protein showed a low similarity with the other NCEDs described in the plant species, with a similarity of 44% to CkNCED1 (ACU86971.1; *Caragana korshinskii*), PsNCED2 (BAC10550.1; *Pisum sativum*) and AhNCED1 (CAE00459.2; *Arachis hypogaea*). The RiNCED2 sequence showed a 100% similarity to FaNCED1 (ADU85829.1) described in *Fragaria* x *ananassa* and 89% to MdNCED1 (AGQ03804.1) from *Malus domestica*. The RiNCED3 showed low homology with the other sequences, showing a 37% = similarity to CcNCED6 (XP006420602.2) from *Citrus clementina* and CisNCED6 (XP006489810.2) from *Citrus sinensis* (Figure 8). 

At the same time, the deduced protein sequences show a length of 210, 190, 216, 218, 174, 160, 192, 184, and 213 amino acid residues for RiPYL1, RiPYL2, RiPYL3, RiPYL4, RiPYL5, RiPYL6, RiPYL7, RiPYL8, and RiPYL9, respectively (Appendix A). Most of the RiPYL sequences identified in raspberry had similarities with the FvPYL sequences described in *Fragaria vesca* (Figure 9). Thus, the RiPYL1, RiPYL2, RiPYL3, and RiPYL9, RiPYL4, RiPLY5, RiPYL6, and RiPYL8 sequences showed a 67%, 99%, 87%, 97%, 99%, 90%, and 93% similarity to FvPYL3 (XP004300241.1FvPYL2 (XP004291031.1), FvPYL4 (XP004302617.1), FvPYL6 (XP_004308958.1), FvPYL13a (XP004293952.1), FvPYL13c (XP011460608.1), and FvPYL7 (XP_004306686.1) of *Fragaria vesca*, respectively. In turn, the RiPYL7 sequence showed a 37% similarity to MdPYL3 (XP008380616.1) and MdPYL9 (XP008376697.2) from *M. domestica* (Figure 9). 

### 2.6. Expression of RiNCED and RiPYL Genes

Transcriptomic data were retrieved from the “*Rubus idaeus* Heritage transcripts v1.0” database [13] available in the genome data base for Rosaceae [29] to explore the differentially expressed genes (DEGs) in the flower, green fruit, and pink fruit. The analysis of expression patterns by FPKM showed upregulated DEGs at the P stage (Figure 10). In this transcriptomic database, *RiNCED2* and *RiPYL9* were not identified. Most of the identified genes have a higher expression in flowers and pink stage and low expression at the G stage such as *RiNCED1* and *RiPYL1-6*, while *RiNCED3* and *RiPYL8* present higher expression at the P stage in comparison to green fruits and flowers (Figure 10).

The RT-qPCR analysis showed that *RiNCED1, RiPYL1,* and *RiPYL8* transcripts increased their expression during development, with the highest expression at the R and OR stages (Figure 11). A significant increase in the *RiNCED1* expression was observed in drupelets at the OR stage in both seasons. In the receptacle, a significant increase in *RiNCED1* expression was observed at the OR stage for the 2020 season and at the R stage for the 2021 season compared to the LG stage (Figure 11). 

Although *RiPYL1* expression tends to increase during ripening in both tissues and seasons, a significantly higher *RiPYL1* transcript level at the OR stage was observed in drupelets in both seasons and receptacles in the 2021 season (Figure 11). Similar to the *RiPYL1* gene, the *RiPYL8* gene expression shows a tendency to increase during ripening in both tissues and seasons, showing a significant increase in drupelets at the R and OR stage in both seasons and in receptacles at the OR stage for the 2021 season (Figure 11). 

However, the standardized components describe a correlation between the ABA content and gene expression variables (Figure 12). In the 2020 season, a low correlation was observed, being positive for the ABA content and *RiNCED1* expression in the receptacle (0.43) and the ABA content and *RiPYL8* expression in the drupelet (0.54). However, in the 2021 season, the ABA content showed a positive correlation with the three studied genes in both tissues, i.e., 0.67 with *RiNCED1*, 0.63 with *RiNPYL1*, and 0.67 with *RiPYL8* in drupelets; and 0.63 with *RiNCED1*, 0.51 with *RiNPYL1*, and 0.47 with *RiPYL8* in the receptacle (Figure 12). The significant increase of the *RiPYL1* and *RiPYL8* gene expression in drupelets at the R and OR stages suggest that the ABA signaling plays an important role in the ripening of raspberry. In addition, the results suggest a correlation between the ABA content and the expression of *RiNCED1*, *RiPYL1,* and *RiPYL8* that depends on the harvest season. 

## 3. Discussion

The first step to take to understand the regulatory role of hormones during raspberry ripening is to determine its association with the quality parameters and the expression of the key phytohormone-related genes. Here, we showed that the ABA content and the expression of ABA biosynthesis (9-cis-epoxycarotenoid dioxygenase like/*RiNCED*)- and receptor (PYRs-like receptors/*RiPYL*)-related genes increase with the progression of fruit ripening, irrespective of the harvest seasons. 

During both seasons, the fruit quality parameters show the typical change during raspberry ripening, a decrease in firmness and acidity and an increase in total soluble solids. Firmness drastically decreased during both seasons (Figure 2). Indeed, it has been previously described that ‘Heritage’ shows a sharp decline in fruit firmness throughout the ripening period compared to ‘Santa Clara’, ‘Santa Catalina’, and ‘Santa Teresa’ cultivars [32]. 

Ethylene has also been reported to be involved in some critical processes of non-climacteric fruit ripening, e.g., coloration in grape [33], lignin accumulation in strawberry [34], and softening and detachment of raspberry drupelets [9,35,36,37]. In grape, the ACC content reached 1000-fold higher levels than ethylene production, suggesting that ACC biosynthesis was not limiting [33]. In raspberry, enhanced ethylene production during ripening at various CO_2_ concentrations has been observed in the same location during different seasons [5,9,11] with a partial delay in the decrease of firmness during the postharvest treatment with the ethylene receptor inhibitor 1-methyl cyclopropene (1-MCP, 1600 ppb), with no effects on titratable acidity and total soluble solids [9]. However, in the present study, ACC production was not correlated with the quality parameters (Figure 4). This result observed during the preharvest together with the partial regulation of firmness by ethylene during the postharvest season [9] suggest the involvement of other hormones in the Heritage cultivar and possibly also in other raspberry varieties in general.

In non-climacteric fruit, ABA plays a crucial role during ripening [18,38]. Previously, ABA was reported to be involved in the ripening of strawberry fruit [17,18,39] and has been described as the principal regulator of the onset of ripening of the grape berry, including softening, sugar accumulation, and coloration [40,41,42,43]. In both fruits, the ABA content has been described as low during the early fruit development stages and high at the ripe fruit stages [14,15]. In strawberry, ABA treatment was reported to accelerate fruit softening and increase color and ethylene production [17].

Previous reports in raspberry show that ABA accumulation reaches the maximum level at the ripe stage (26,920 ng/g), with the complete correspondence of a high expression of *RiNCED1* [12]. In the present study, the ABA content was near 7000 ng/g DW in the drupelets at the R stage and was higher in the receptacle at the same stage (between 10,000 to 25,000 ng/g DW depending on the harvest seasons). Similar to the results reported by Yang et al., 2020 [12], our results showed that the ABA content in drupelets is maintained or decreased in the OR stage compared to the content in ripe fruits. On the contrary, the ABA content in the receptacle continued increasing at the OR stage in both seasons (Figure 3). It has been shown that postharvest application of exogenous ABA (1 mM) at the large green stage promotes ABA biosynthesis, the accumulation of the soluble sugar and anthocyanin contents, softening of fruit, and chlorophyll loss in the raspberry fruit [12]. The same authors [12] showed that fluoridone, an ABA biosynthesis inhibitor, diminished the ABA content, and consequently, delayed fruit softening, and reduced the chlorophyll content in raspberry fruit. In this study, the correlation matrix shows a positive correlation between the TSS and ABA content, a negative correlation between firmness and the ABA content, as well as and between the TA and ABA content in both seasons (Figure 4). The previous reports of Yang [12] and our results imply that ABA played an important role as a regulator of raspberry ripening. However, treatments that inhibit ABA biosynthesis and signaling should be carried out in fruit attached to the plant to accurately determine the role of this hormone and its possible relationship with other compensatory mechanisms during fruit development.

Regarding the ABA metabolism, the *NCED* gene encodes an enzyme that catalyzes the conversion of violaxanthin or neoxanthin into xanthoxin and is a key regulatory component for the ABA biosynthesis pathway, being widely reported during the ripening of climacteric and non-climacteric fruit [15,19,20,21,44]. Although three *NCED* genes have been identified in strawberry (i.e., *FaNCED1*, *FaNCED2*, and *FaNCED3*), *FaNCED1* genes are closely related to ABA biosynthesis and the strawberry ripening process [15,19]. In this study, three genes of the *RiNCED* family (Figure 5; Appendix A) were identified based on the genomic information of *Rubus idaeus* [28]. The *RiNCED* family exhibits a high degree of conservation according to their gene structure and the motif profile of the deduced proteins (Figure 5). The promoter analysis of *RiNCED* genes shows the presence of 15 ABA cis-elements for *RiNCED3*, 9 for *RiNCED1*, and 6 for *RiNCED2*. Also, the three promoter sequences present cis-elements in response to auxin, jasmonates, and salicylic acid. Only the promoter of *RiNCED1* shows cis-elements that respond to ethylene; conversely, RiNCED2 and RiNCED3 promoters showed no elements of the ethylene response (Figure 7). Although the analysis of the promoters indicates a potential regulation of *NCED* expression by other phytohormones, further studies must be carried out to demonstrate this relationship between both hormones in raspberry. In peach (*Prunus persica*), a yeast one-hybrid assay suggested that the nuclear-localized PpERF3 might bind to the promoters of *PpNCED2/3* [44]. Also, promoter-GUS assays and transient expression analyses of PpERF3 increased the expression of *PpNCED2/3*. Both results suggested that ethylene promotes ABA biosynthesis by regulating the expression of *PpNCED2/3* mediated by PpERF3 [44]. In contrast, the jasmonate signaling pathway could have an antagonistic effect on ABA biosynthesis in strawberry (*Fragaria* × *ananassa*), since it has been reported that the ABA content decreased together with the downregulation of *FaNCED1* after methyl jasmonate (MeJA) treatment for five days in strawberry fruit [45]. 

In this study, the phylogenetic analysis showed that the *RiNCED* gene family comprised three subfamilies—group I, II, and V—of the five NCED groups described in plants (Figure 8). RiNCED2 was the deduced protein sequence with a major homology to the other NCED described in fruits, showing a 100% similarity to FaNCED1 (ADU85829.1) described in *F.* × *ananassa* and 89% to MdNCED1 (AGQ03804.1) from *M. domestica*. 

Regarding ABA signaling, the PYL ABA receptor gene family has been described in fruits such as tomato [24], strawberry [25], and grape [26]. However, no information has been reported in raspberry until now. In this study, the genomic information shows a total of nine *RiPYL* members (Figure 6; Appendix A) with differences in the exon and intron distribution and content (Figure 6A). Likewise, nine members of *FaPYL* in *F.* × *ananassa* [25] have been described. Although differences were observed in the presence of introns between the nine copies of the *RiPYL* genes, the deduced proteins showed that six conserved motifs were similarly distributed (Figure 6B), indicating the low variability in the final protein sequence of this gene family. In apple, 13 *PYL* genes have been identified classified into four groups according to the structural features of the amino acid residue sequence and they contained at least three cis-elements related to the hormonal response and stress in their promoter regions [46]. For example, the *MdPYL7* promoter had cis-elements, including abscisic acid-responsive elements (ABRE), salicylic acid-responsive elements (TCA-element), and MeJA-responsive elements (CGTCA-motif, TGACG-motif). The *MdPYL13* promoter region also contained ethylene-responsive elements (ERE) [46]. Our phylogenetic analysis showed that *RiPYL* genes were distributed within the three groups described in plants (Figure 9), and most of the deduced proteins have a similarity of over 67%, with the FvPYL sequences described in *F. vesca*. The main similarity (99%) was between the RiPYL2 and FvPYL2 sequences, and between the RiPYL5 and FvPYL13a sequences. In *F.* × *ananassa*, *FaPYL2* showed the highest transcript level, and its transcript abundance increased rapidly at the onset of red coloration [25]. However, a minor similarity (37%) was observed between the RiPYL7 and MdPYL3/9 sequences described in apple (Figure 9). In this fruit, the *MdPYL9* gene was significantly induced by drought treatment, and its over-expression conferred enhanced tolerance to drought stress in transgenic apple plants [23]. 

In this study, most of the identified genes in the transcriptomic data have a higher expression at the pink stage (Figure 10). The RT-qPCR analysis showed that the expression of the *RiNCED1*, *RiPYL1*, and *RiPYL8* transcripts increased during development, with the highest expression levels at the R and OR stages (Figure 11). In grape and peach, *VvNCED1* and *PpNCED1* were highly expressed at the onset of the ripening of the fruit, preceding the ABA accumulation, and were particularly necessary for the increase in ethylene production during the ripening of peach [47]. The respective decrease and increase of the *RiNCED1* expression after fluoridone and ABA application [12], together with our results, suggest that ABA biosynthesis is necessary for the onset of the ripening and persists until the senescence stages in the receptacle and drupelet tissues.

ABA signaling has been little studied in relation to fruit ripening thus far. In this study, the expression of both genes related to ABA signaling, *RiPYL1,* and *RiPYL8*, increased during ripening in both tissues and seasons, showing marked differences in drupelets (Figure 11). In strawberry, it has been shown that during the de-greening from the large green to white stages, the transcript levels of *FaPYL2/4/8/9* decreased rapidly and increased during coloration from the pink to red stages [25].

Our correlation analyses obtained with the data from two seasons indicated that the feedback of ABA on the expression of the *RiNCED1*, *RiPYL1*, and *RiPPYL8* genes could depend on the season (Figure 12); however, it is important to highlight that the expression of *RiNCED1* maintains a positive correlation with the ABA content in the receptacle during both seasons, as well as the expression of *RiPYL8* in drupelets. 

## 4. Materials and Methods

### 4.1. Plant Material

The raspberry (*R. idaeus* ‘Heritage’) fruit was collected from commercial orchards located in Casa Blanca (33°20′39″ S; 71°22′11″ W; 247 masl), Chile, during two harvest seasons in 2020 and 2021. Fruit attached to the receptacle and with peduncle were harvested and sorted by size and color as large green (LG), white (W), pink (P), red (R), and overripe fruit (OR) [9,27]. Immediately after harvest, half of the collected fruit was frozen in liquid nitrogen for ABA and ACC determination, and RNA isolation. The other half was used to determine the quality and physiological parameters.

### 4.2. Fruit Quality and Physiological Measurements

Five independent replicates of 15 intact fruit attached to the receptacle per developmental stage were used for quality assessments. The firmness of the fruit was determined using the Firm Tech II equipment (BioWorks Inc., Wamego, KS, USA) and expressed as Newton (N) [9]. Eight grams of fruit tissue (drupelets) from each replicate were homogenized in a mortar. The juice was analyzed for total soluble solids (TSS) using a refractometer (ATAGO, Tokyo, Japan), expressed as °Brix, and the titratable acidity (TA) was expressed as the percentage of citric acid per 100 g of the fresh weight (FW) of fruit. 

### 4.3. Determination of ABA and ACC Contents

The ABA and ACC contents during the ripening of raspberry were determined according to Müller and Munné-Bosh (2011) [48]. First, fruit with a peduncle was collected and immediately frozen in liquid nitrogen. Then, eight independent samples of four fruit were freeze-dried, drupelets and receptacles were separated, and each tissue was powdered in a mortar. Then, 100 mg of samples were extracted after adding internal standards. d6-ABA and d4-ACC were added as the internal standards. After centrifugation (10,000 rpm for 15 min at 4 °C), the supernatant was collected, the pellet was reextracted with 0.5 mL of the extraction solvent, and the extraction was repeated three times. Then, supernatants were combined and dried under a nitrogen stream, re-dissolved in 300 μL of methanol, centrifuged (10,000 rpm for 5 min), and filtered through a 0.22 μm PTFE filter (Waters, Milford, MA, USA). ABA and ACC were determined by UPLC/ESI-MS/ MS. The quantification was carried out by creating a calibration curve including unlabeled analyte compounds (ABA and ACC standards) ranging from 0.05 to 1000 ng ml^−1^. Calibration curves for each analyte were generated using the Analyst™ software (Applied Biosystems, Inc., California, USA).

### 4.4. Identification of RiNCED and RiPYL Genes in the Raspberry Genome

The *RiNCED* and *RiPYL* genes were identified in the draft genome published by Wight et al., 2019 [28] using the Geneious Prime software [49]. Sequences of raspberry *RiNCED* and *RiPYL* genes previously identified in the transcriptomes published by Travisany et al., 2019 [13] and Wight et al., 2019 [28] were used as query sequences for a Blast search. The obtained genomic sequences were analyzed with the Augustus server [50] for gene prediction, intron-exon structure, coding sequence, and protein sequence.

### 4.5. Gene Structure and Promoter Analysis of RiNCED and RiPYL Genes

The intron-exon structure of the *RiNCED* and *RiPYL* genes was displayed by the Gene Structure Display Server 2.0 [51,52]

Promoter analysis was performed using sequences of 2000 bp upstream of the transcriptional start site obtained from the draft genome and analyzing them with the Plant-CARE software [53] for the identification of the cis-elements related to the phytohormone response.

### 4.6. Characterization of RiNCED- and RiPYL-Deduced Protein Sequences 

The phylogenetic analysis of RiNCED- and RiPYL-deduced proteins was carried out using the Maximum Likelihood Method with a bootstrap value of 5000 replicates with the MEGA-X software [54] using the deduced protein of other plant species (Appendix A). The conserved motifs were identified by Multiple EM for the Motif Elicitation tool (MEME version 5.4.1) [55,56]. 

### 4.7. Identification of the Differentially Expressed RiNCED and RiPYL Genes in the Transcriptomic Data

To identify the differentially expressed *RiNCED* and *RiPYL* genes in the flower, green, and pink fruit stages, a search was realized on a database of differentially expressed genes (DEGs) “*Rubus idaeus* Heritage transcripts v1.0” [13] available in the Genome Data Base for Rosaceae [29]. For the search, a Blastn was performed on the Genome Data Base for Rosaceae page [57], and the coding sequences (CDS) of the *RiNCED* and *RiPYL* genes were used as queries. Then, the values of fragments per kilobase of transcript per million mapped fragments (FPKM) were determined for each identified sequence. These values were expressed as log_10_ FPKM^+1^ to plot a heatmap later using the CummeRbund R package [58].

### 4.8. RNA Isolation and Reverse Transcription Quatitative PCR (RT-qPCR) Analysis

Five fruits were powdered with liquid nitrogen in a mortal and 100 mg of powder was used for the total RNA isolation from drupelets, and receptacle of each ripening stage using the RNAqueous^®^Kit (ThermoFisher Scientific Inc. Waltham, MA, USA). DNase treatment was performed using DNase I (Thermo Fisher Scientific Inc.). The first-strand cDNA was generated using High-Capacity cDNA Reverse Transcription Kits (ThermoFisher Scientific Inc.). Five individual extractions for each sampling were performed. Specific forward and reverse primers were designed with high stringency using Primer3Plus [59]. The primers for the housekeeping and candidate genes are detailed in Appendix A. The expression of the candidate genes was carried out using the KAPA SYBR^®^FAST qPCR Kit (Kapa Biosystems, Wilmington, MA, USA) in the PikoReal^®^ Real-Time PCR System (ThermoFisher Scientific Inc.) according to Delgado et al., 2018 [60] with minor modifications. Each reaction was performed in triplicate and normalized against the expression level of the *Ri18S* gene [13]; additionally, negative water control was included. The results were calculated using the 2^−ΔΔCT^ method [61] using fruit at the LG stage as the calibrator sample and assigned a nominal value of 1.

### 4.9. Statistical Analysis

The data for each season were analyzed using the R Statistical Software [62]. For the ABA and ACC content, firmness of TSS and TA, and relative expression, normality was corroborated using the Shapiro-Wilk normality test. Based on this, it was decided which comparison test should be used: the non-parametric Wilcoxon rank sum for one-sampled data and Mann-Whitney for two-sampled data; otherwise the t-test was used. The resultant *p*-values were adjusted using Bonferroni’s method. Also, the principal component analysis was used to analyze the grouped data by year. The ABA and ACC content, firmness of TSS and TA, and the relative expression values obtained during development were correlated using the Kendall method. The data were visualized using the R package ggstatsplot [63] and rstatix [64].

## 5. Conclusions

In this study, the ABA content showed a higher increase during the ripening of raspberry, including the receptacle and drupelet tissues, during two seasons. Furthermore, this increase was correlated with the change in the fruits’ quality parameters (i.e., firmness lost, increase of soluble solids). In addition, we identified and characterized the main ABA biosynthesis (*RiNCED*)- and ABA receptor (*RiPYL*)-genes, which showed a positive correlation between their expression levels and the progress of the ripening of the raspberry and the potential differential expression between fruit tissues. Further studies, including the functional characterization of these key ABA pathway-related genes, could improve the understanding of the role of this hormone in the raspberry fruit and the differences in ABA signaling between drupelets and receptacles.

## Figures and Tables

**Figure 1 plants-12-01882-f001:**
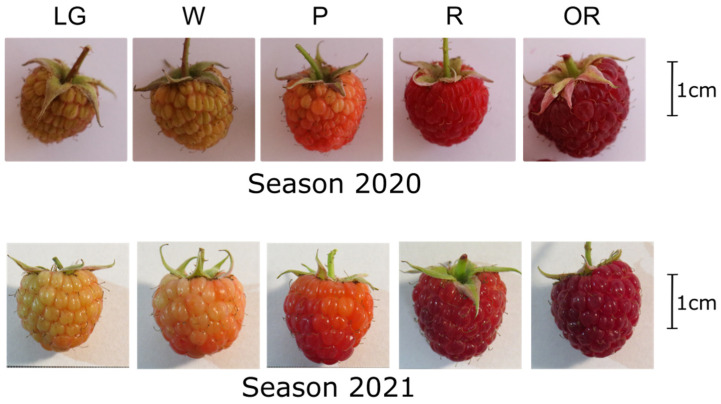
Developmental stages of raspberry (*R. idaeus* ‘Heritage’) fruit during 2020 and 2021 seasons. The fruit was harvested and sorted by size and color as follows: large green (LG), white (W), pink (P), red (R), and overripe (OR) fruit.

**Figure 2 plants-12-01882-f002:**
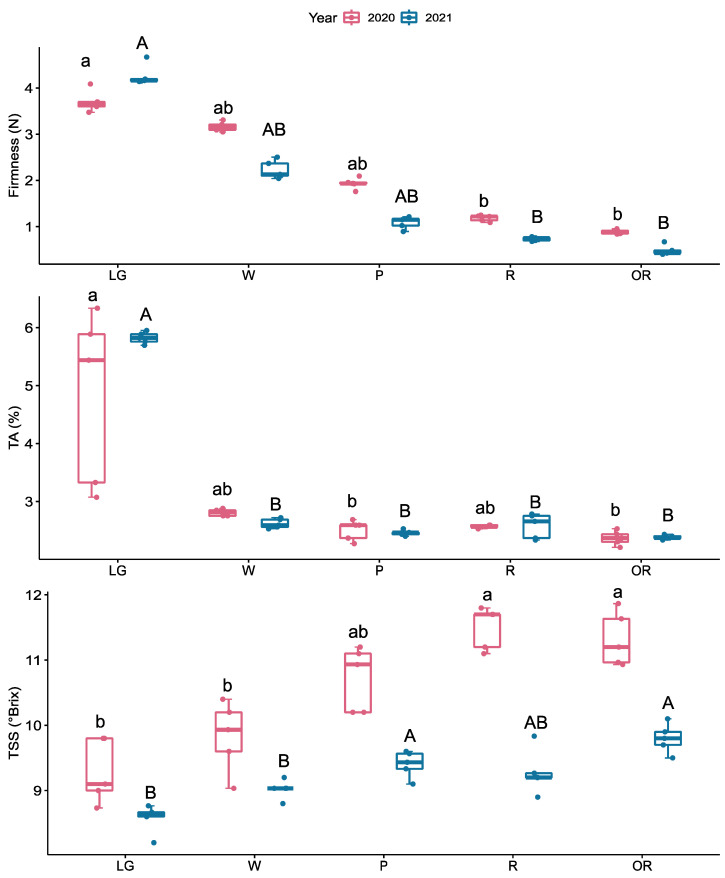
Quality parameters during the development of raspberry (*R. idaeus* ‘Heritage’) fruit. Firmness (N) was determined in the whole fruit, and total soluble solids content (TSS, °Brix) and titratable acidity (TA, %) were determined in drupelets. The fruit was harvested and sorted by size and color as follows: large green (LG), white (W), pink (P), red (R), and overripe (OR) fruit stage. Data represent the means ± S.D. from five sample units (each containing ten fruits) by the development stage. Significant differences are indicated (*p* ≤ 0.05) between developmental stages as lowercase for 2020 and uppercase for 2021. Boxplots show each group’s distribution (developmental stages), and the black line (central value in the box) indicates the median value for non-parametric analysis and points outside the boxplots show outlier values.

**Figure 3 plants-12-01882-f003:**
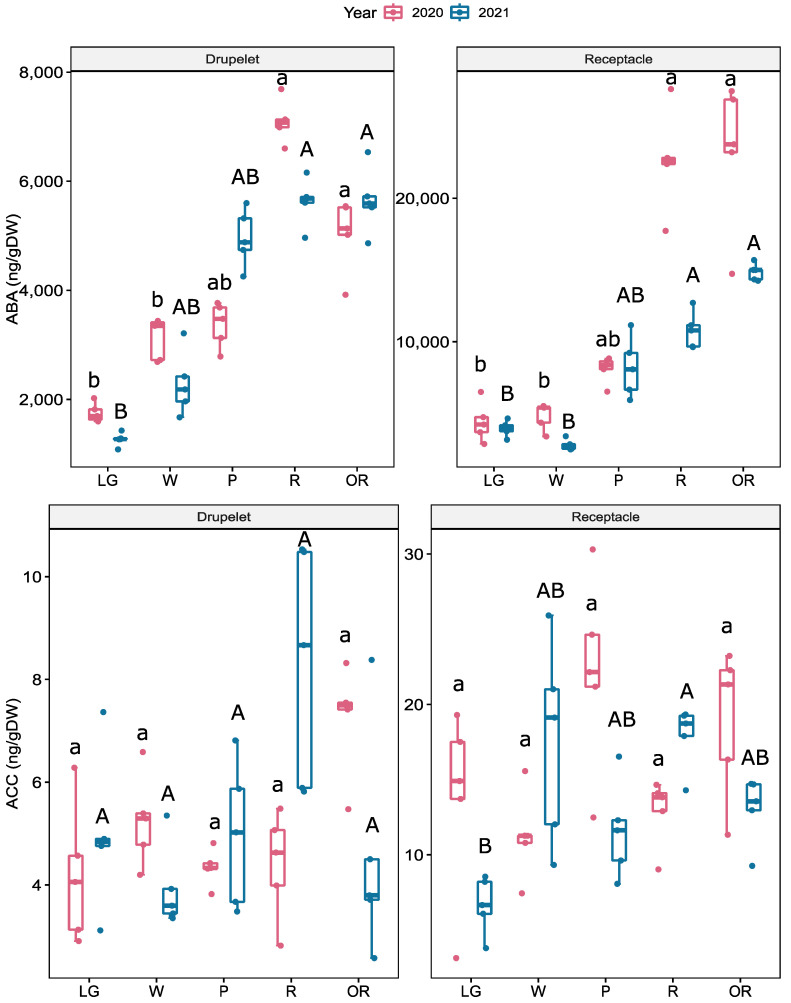
ABA and ACC contents during the development of raspberry (*R. idaeus* ‘Heritage’) fruit. The fruit was harvested in the 2020 and 2021 seasons and sorted by size and color as follows: large green (LG), white (W), pink (P), red (R), and overripe (OR) fruit. ABA and ACC contents (expressed as ng/g DW) were determined in drupelets and receptacles from each stage using five independent replicates (each containing ten fruits). Significant differences are indicated (*p* ≤ 0.05) between developmental stages as lowercase for 2020 and uppercase for 2021. The boxplots show each group’s distribution (developmental stages), the black line (the central value in the box) indicates the median value for non-parametric analysis, and the points outside the boxplots show outlier values. Otherwise, the central value represents the mean value. DW: dry weight.

**Figure 4 plants-12-01882-f004:**
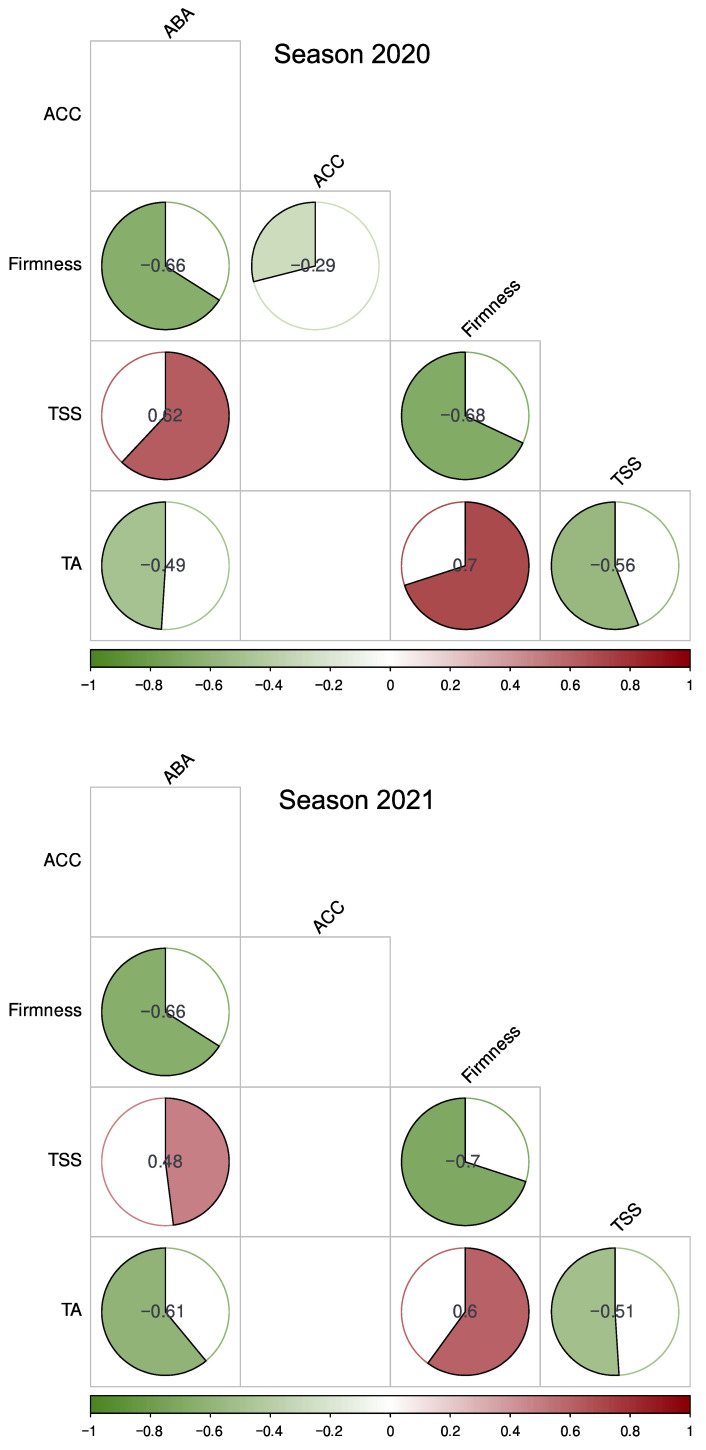
Correlation matrix of variables in drupelets during the development of the raspberry (*R. idaeus* ‘Heritage’) fruit in two seasons. The correlation was carried out by the Kendall method. ABA: abscisic acid content; ACC: 1-aminocyclopropane-1-carboxylic acid content; TSS: total soluble solids content; and TA: titratable acidity. Significant correlation values (*p*-value ≤ 0.05) are shown inside the circles.

**Figure 5 plants-12-01882-f005:**
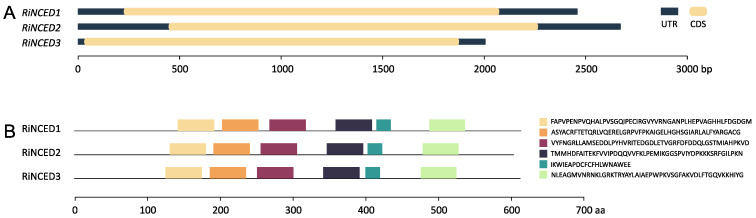
Gene structure of *RiNCED*s and conserved protein motifs of deduced proteins. (**A**) Gene structures for *RiNCED*. (**B**) Motif profile of RiNCED deduced proteins. The six motifs are displayed in different colored boxes. CDS: coding sequence; UTR: untranslated region.

**Figure 6 plants-12-01882-f006:**
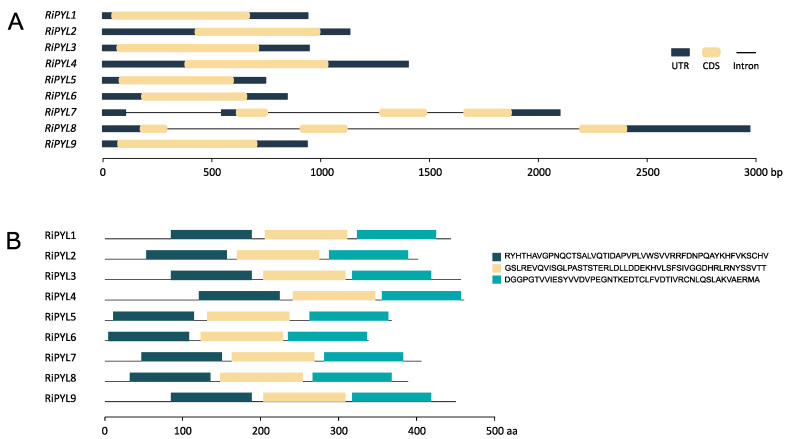
Gene structure of *RiPYLs* and conserved protein motifs of deduced proteins. (**A**) Exon-intron structures for *RiPYL* genes. (**B**) Motif profile of RiPYL deduced proteins. The three motifs are displayed in different colored boxes. CDS: coding sequence; UTR: untranslated region.

**Figure 7 plants-12-01882-f007:**
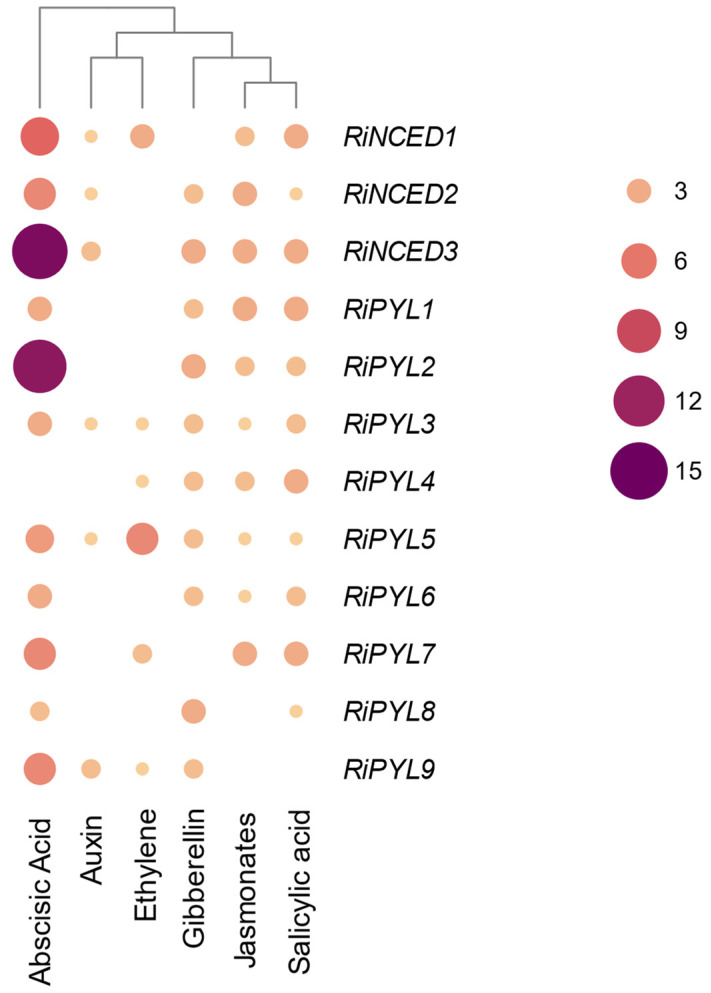
Promoter analysis of *RiNCED* and *RiPY*L genes identified in *R. idaeus* genome. The figure represents the number of cis-elements that respond to phytohormones in the promoter regions for each sequence identified in the genome draft of raspberry.

**Figure 8 plants-12-01882-f008:**
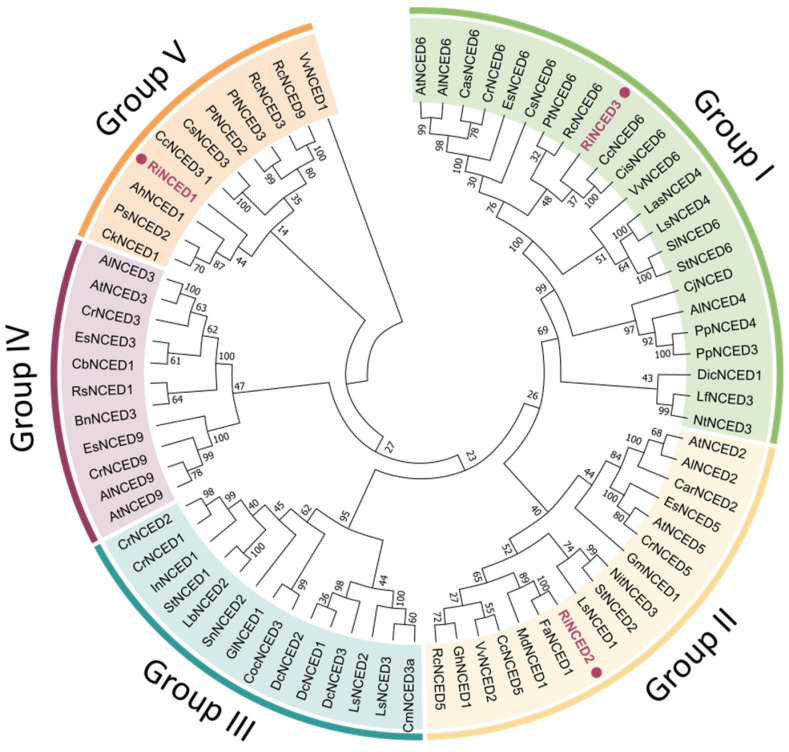
Phylogenetic tree of NCED deduced proteins from *R. idaeus* and other plant species. The phylogenetic tree was constructed based on the full-length sequences of the RiNCED-deduced proteins using the MEGA-X software. Purple dots are NCED deduced protein identified in raspberry.

**Figure 9 plants-12-01882-f009:**
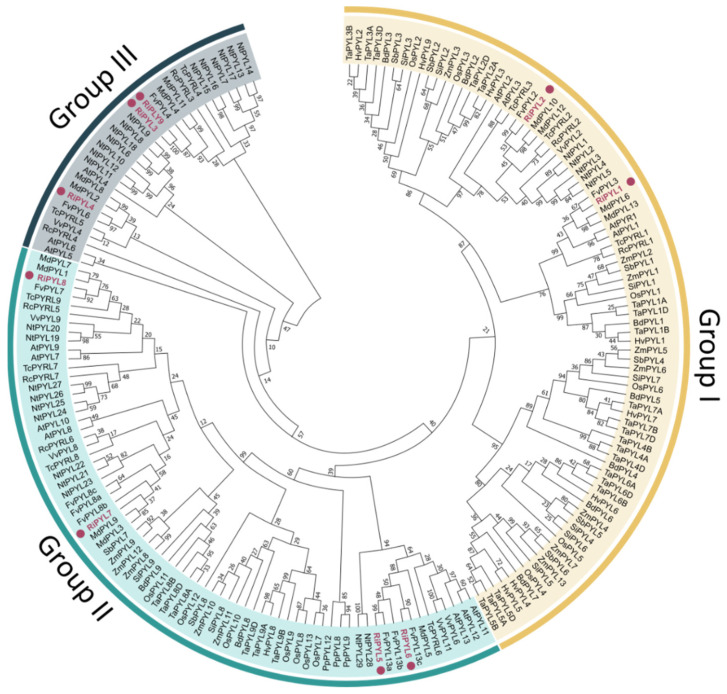
Phylogenetic tree of PYL deduced proteins from *R. idaeus* and other plant species. The phylogenetic tree was constructed based on the full-length sequences of the RiPYL deduced proteins using the MEGA-X software. Purple dots are PYL deduced protein identified in raspberry.

**Figure 10 plants-12-01882-f010:**
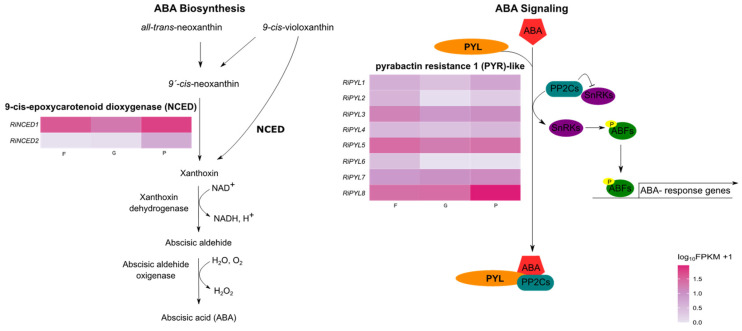
*RiNCED* and *RiPYL* genes differentially expressed in database “*R. idaeus* Heritage transcripts v1.0. NCED and PYL are included in the ABA biosynthesis and signaling pathways, respectively. Gene expression was calculated as FPKM derived from flower (F), green fruit (G), and pink fruit (P) RNA-Seq data [13,29]. ABA biosynthesis pathway modified from Han et al., (2004) [30]; ABA signaling pathway modified from Yoon et al., (2020) [31].

**Figure 11 plants-12-01882-f011:**
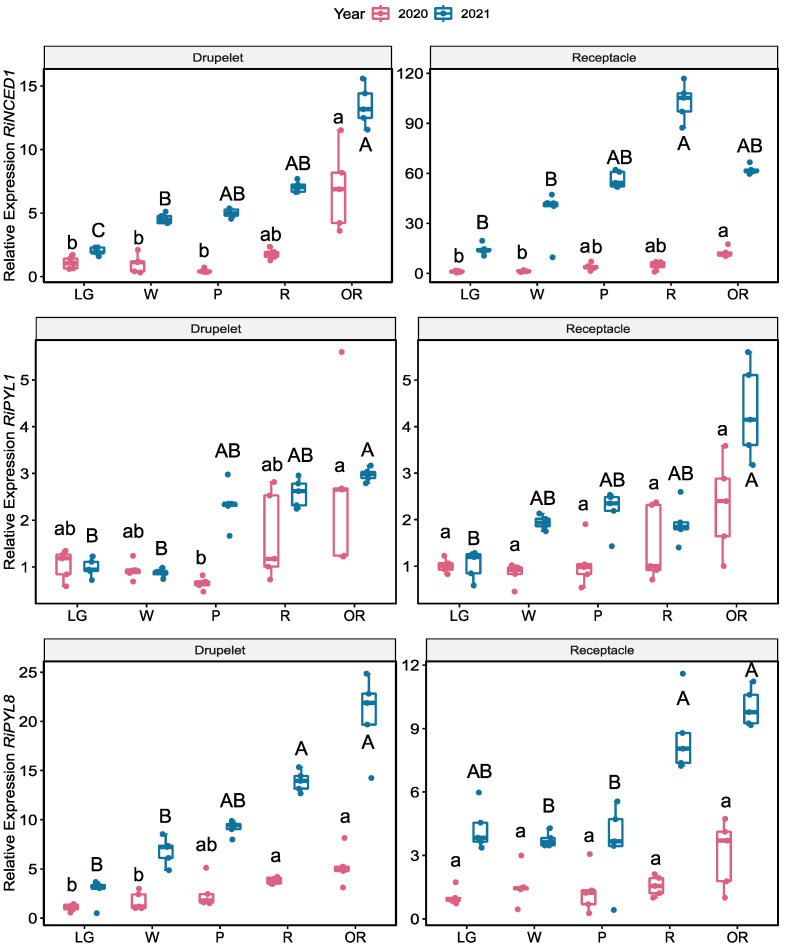
Expression profiles of the *RiNCED1*, *RiPYL1,* and *RiPYL8* genes during the development of raspberry (*R. idaeus* ‘Heritage’) fruit. The fruit was harvested and sorted by size and color as follows: large green (LG), white (W), pink (P), red (R), and overripe (OR) fruit stages during the 2020 and 2021 seasons. Relative expression was determined in drupelets and receptacles from each stage using five independent sample extractions. *Ri18S* was used as the normalizer gene. Significant differences are indicated (*p* ≤ 0.05) between the developmental stages as lowercase for 2020 and uppercase for 2021. Boxplots show each group’s distribution (developmental stages), and the black line (central value in the box) indicates the median value for non-parametric analysis and points outside the boxplots show outlier values. Otherwise, the central value represents the mean value.

**Figure 12 plants-12-01882-f012:**
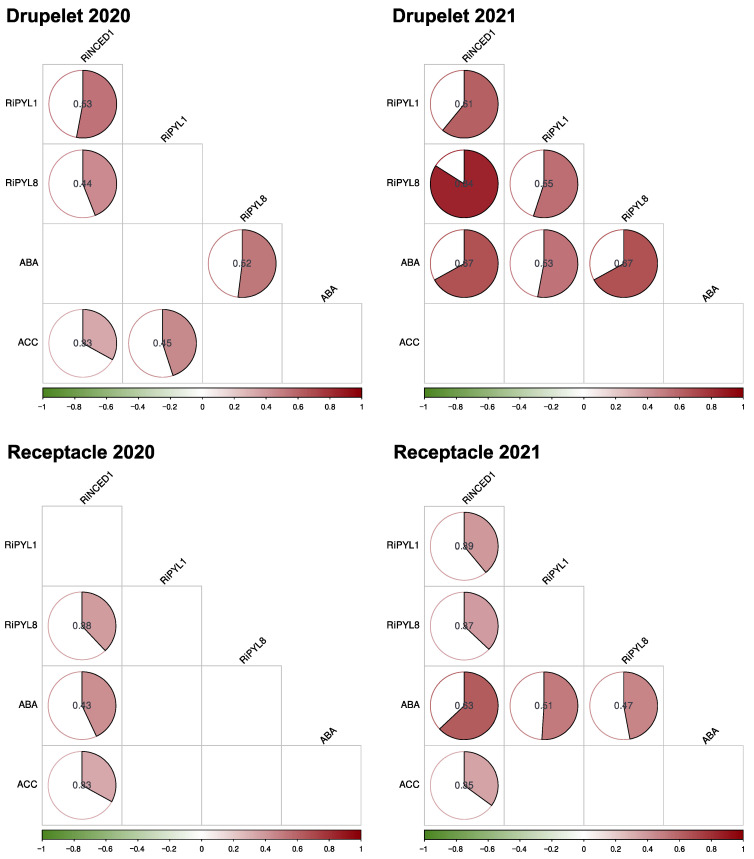
Correlation matrix of variables during raspberry (*R. idaeus* ‘Heritage’) fruit development in two seasons. The correlation was carried out by the Kendall method. ABA: abscisic acid content; ACC: 1-aminocyclopropane-1-carboxylate acid; *RiNCED1*: expression of the raspberry NCED1 ABA biosynthesis gene; *RiPYL1*: expression of the raspberry PYL1 ABA receptor gene; *RiPYL8*: expression of the raspberry PYL8 ABA receptor gene. Significant correlation values (*p*-value ≤ 0.05) are showed inside the circles.

## Data Availability

Not applicable.

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
