# Peer review of "Abscisic Acid Synthesis and Signaling during the Ripening of Raspberry (Rubus idaeus ‘Heritage’) Fruit"

_plants, 2023, doi:10.3390/plants12091882_

Round 1
Reviewer 1 Report
This manuscript found the phenomenon that ABA content was significant increased during the ripening of raspberry, with higher content in the receptacle of ripe and overripe stages. Moreover, the expression level of ABA biosynthesis- and ABA receptor-related genes increased from the large green stage to the full-ripe stage, specifically characterized by a higher expression of RiNCED1 in the receptacle tissue. However, there are some concerns need to be addressed before publication.
1. The manuscript just found a consistent concomitant increase in the ABA content during raspberry ripening. However, that whether ABA signaling is important for raspberry ripening is still unknown. To some content, the results was not helpful for the solution of the question that “the role of other hormones, such as abscisic acid (ABA), in raspberry ripening is unclear”.
2. A RNAi or knockout of RiNCED1 is needed to make clear that whether ABA is directly related to raspberry ripening.
Author Response
Response to Reviewer 1
This manuscript found the phenomenon that ABA content was significant increased during the ripening of raspberry, with higher content in the receptacle of ripe and overripe stages. Moreover, the expression level of ABA biosynthesis- and ABA receptor-related genes increased from the large green stage to the full-ripe stage, specifically characterized by a higher expression of RiNCED1 in the receptacle tissue. However, there are some concerns need to be addressed before publication.
1) The manuscript just found a consistent concomitant increase in the ABA content during raspberry ripening. However, that whether ABA signaling is important for raspberry ripening is still unknown. To some content, the results was not helpful for the solution of the question that “the role of other hormones, such as abscisic acid (ABA), in raspberry ripening is unclear”.
R: We agree with reviewer 1 that the question of ABA's role in ripening still needs to be answered. Since there is no further information regarding the hormone contents in raspberry fruit, we need to establish a basis for deciphering the role of ABA during ripening. This is the first approach to characterize the ABA pathway (i.e., synthesis and signaling) and its relationships with ripening parameters and fruit quality. We are undergoing hormonal treatments to answer this, but it will take time. Therefore, we modified some paragraphs and were more cautious about the current scope of our results. In addition, we are determining other hormonal metabolites, and ABA is the only one consistent with ripening in different seasons and localities.
2) A RNAi or knockout of RiNCED1 is needed to make clear that whether ABA is directly related to raspberry ripening.
R: We agree with reviewer 1 that knockout of RiNCED1 is an excellent alternative to determine the relationship of ABA biosynthesis with raspberry ripening. Indeed, we are setting up transient expression assays to determine the role of some candidate genes in raspberry ripening, and we keep in mind to perform gene editing to get NCED1 knockout plants. Nevertheless, we think that the present research is a first good step into characterizing the role of ABA in raspberry ripening.
The authors thank the Reviewer comments
Reviewer 2 Report
Authors investigated that the raspberry firmness, acidity, color, ABA content and related gene expressions, and found that ABA could participate in the process of raspberry ripening. The findings are interesting, while the ways of data presented must be improved.
1. The data of firmness, total soluble solids, titratable acidity should be presented in the manuscript, not in the supplementary data.
2. All the data can be presented by column diagrams (data of 2020 and 2021 can be shown in one figure) to decrease the number of figures.
3. correlation matrix can be shown in tables with the exact number and significance in it.
4. There is a mistake in Fig. 6, it. is not CDS in the gene.
5. I suggest the author to have a pathway of ABA signaling and synthesis, and put the gene expression data in it, for example Liu et al. (2020).
Author Response
Response to reviewer 2
Authors investigated that the raspberry firmness, acidity, color, ABA content and related gene expressions, and found that ABA could participate in the process of raspberry ripening. The findings are interesting, while the ways of data presented must be improved.
1) The data of firmness, total soluble solids, titratable acidity should be presented in the manuscript, not in the supplementary data.
R: Thank you for this suggestion. We changed this data to a new figure in the main text (please, see Fig. 2 in the new version).
2) All the data can be presented by column diagrams (data of 2020 and 2021 can be shown in one figure) to decrease the number of figures.
R: We agree. We put data from both years in the same figure (see Fig. 2, Fig. 3, and Fig. 11 in the new version) maintaining the visualization of statistical analysis by season.
3) correlation matrix can be shown in tables with the exact number and significance in it.
R: As the reviewer suggests, we improved this figure (Fig. 4 in the new version) with the correlation values indicated inside the circles and indicate in the legend that the correlations are significant at 95%.
4) There is a mistake in Fig. 6, it. is not CDS in the gene.
R: As the reviewer suggests, we improved Fig. 6. RiNCED genes do not have introns, and the legend was changed accordingly.
5) I suggest the author to have a pathway of ABA signaling and synthesis, and put the gene expression data in it, for example Liu et al. (2020).
R: As the reviewer suggested, we improved the Figure of DEGs (Fig. 10 in the new version) in the database "Rubus idaeus Heritage transcripts v1.0, similar to Liu et al., 2020 (https://doi.org/10.1038/s41438-020-00360-7). The heat maps of RiNCEDs and RiPYLs genes were included in the respective pathway.
The authors thank the Reviewer's comments
Round 2
Reviewer 1 Report
The revised version nearly solve all my concerns through alternative description of the results.
Author Response
Response to Reviewer 1:
The revised version nearly solve all my concerns through alternative description of the results.
R: The authors thank the Reviewer's comments
Reviewer 2 Report
The title is not appropriate otherwise it can be accept.
How about "Role of abscisic acid in raspberry during fruit ripening".
Author Response
Response to Reviewer 2:
The title is not appropriate otherwise it can be accept. How about "Role of abscisic acid in raspberry during fruit ripening".
A: We agree with reviewer 2. Therefore, we modified the manuscript title.
The authors thank the Reviewer's comments